# Impact of adult weight management interventions on mental health: a systematic review and meta-analysis protocol

Rebecca A Jones ![ORCID],[1] Emma R Lawlor,[1] Simon J Griffin,[1,2] Esther M F van Sluijs,[1,3] Amy L Ahern[1]

¹MRC Epidemiology Unit, University of Cambridge, Cambridge, UK
²Primary Care Unit, Institute of Public Health, University of Cambridge, Cambridge, UK
³Centre for Diet and Activity Research (CEDAR), University of Cambridge, Cambridge, UK

**Correspondence to**
Rebecca A Jones;
rj397@cam.ac.uk

## ABSTRACT

**Introduction** The effects of interventions targeting weight loss on physical health are well described, yet the evidence for mental health is less clear. It is essential to better understand the impact of weight management interventions on mental health to optimise care and minimise risk of harm. We will assess the effect of behavioural weight management interventions on mental health in adults with overweight and obesity.

**Methods and analysis** The systematic review will follow the Preferred Reporting Items for Systematic Reviews and Meta-Analyses guidance. We will include behavioural weight management interventions with a diet and/or physical activity component focusing on weight loss for adults with a body mass index ≥25 kg/m². Randomised controlled trials (RCTs) and cluster RCTs will be the only eligible study designs. Outcomes of interest will be related to mental health. The following databases were searched from inception to 07 May 2019: MEDLINE, Embase, Cochrane database (CENTRAL), PsycINFO, ASSIA, AMED and CINAHL. The search strategy was based on four concepts: (1) adults, defined as ≥18 years, with overweight/obesity, defined as BMI ≥25kg/m², (2) weight management interventions, (3) mental health outcomes and (4) study design. The search was restricted to English-language published papers, with no other restrictions applied. Two stage screening for eligibility will be completed by two independent reviewers, with two independent reviewers completing data extraction and risk of bias assessment. Data permitting, a random-effects meta-analysis of outcomes, subgroup analyses and meta-regression will be conducted. If not appropriate, narrative synthesis and 'levels of evidence' assessment will be completed.

**Ethics and dissemination** Ethical approval is not required as primary data will not be collected. The completed systematic review will be disseminated in a peer-reviewed journal, at conferences and contribute towards the lead author's PhD thesis.

**PROSPERO registration number** CRD42019131659.

## INTRODUCTION
### Rationale

Overweight and obesity are strongly associated with reduced physical health, including

### Strengths and limitations of this study

► The systematic review and meta-analysis will include only randomised controlled trials, offering the highest level of evidence.
► A broad array of mental health outcomes, including mood, stress and depression, will be included in the review.
► A comprehensive search strategy will be used in a large number of databases to maximise the identification of all eligible studies.
► Data permitting, subgroup analysis will be conducted to identify intervention or participant characteristics associated with increased effectiveness.
► High heterogeneity is anticipated across studies, which may increase the difficulties in interpreting a meta-analysis.

a greater risk of cardiovascular disease, type 2 diabetes, stroke and some cancers (including endometrial, oesophageal and kidney cancer).[1–3] Consequently, individuals with overweight and obesity experience greater all-cause mortality and reduced health-related quality of life.[4 5] Research reports a bidirectional association between obesity and mental health, with those with overweight and obese more likely to have poor mental health and those with mental ill health at greater risk of weight gain, and consequently, obesity.[6–10] Many researchers have reported improvements in mental health outcomes with weight loss,[11–15] however, there has been concern expressed that weight management interventions advocating dietary restriction may contribute to disordered eating and worsen mental health.[16 17] It is essential to better understand the impact of weight management interventions on mental health to optimise care and minimise risk of harm.

Research investigating the relationship between obesity and mental health is

BMJ

increasingly considering mental health as a symptom continuum. The symptom continuum appreciates that individuals can experience one or more symptoms of mental illness without meeting diagnostic criteria for mental illness.[18 19] Considering mental health as a continuum is associated with reduced stigma and improved attitudes towards mental health, highlighting the benefits of broadening the definition of mental health.[19 20] This review will embrace a continuum-based definition of mental health allowing the investigation of a broader range of outcomes from stress, self-esteem and affect, to symptoms of clinically diagnosed disorders, such as depression and anxiety.

While there is clear evidence that weight loss interventions improve physical health, the evidence that they enhance mental health is less clear. Some studies suggest that a focus on weight control can increase stigma and exacerbate symptoms of psychological distress,[21] particularly if goals are not met or if other aspects of life do not change with weight loss.[22] Qualitative research has suggested that there is inadequate support for mental health in obesity management interventions,[23] and a systematic review published in 2014 concluded that weight loss may be associated with improved physical health, but not mental health.[24] Conversely, Fabricatore *et al*'s review found statistically significant reductions in depressive symptoms with intentional weight loss trials, although it reported no relationship between weight change and depression,[9] and Lasikiewicz *et al*'s review reported weight management interventions to be associated with improvements in multiple mental health outcomes including self-esteem, body image, quality of life and depressive symptoms.[25]

Previous reviews highlight the breadth of mental health outcomes that could be affected by participation in weight management programmes. However, the majority of reviews focus on a limited range of outcomes,[9 16 24–28] and the direction of effects is inconsistent across different outcomes and reviews. It is important to generate a comprehensive understanding of the impact of weight management programmes on mental health as the benefits of improvements in one domain may be undermined by negative impacts on other domains. Previous reviews have also excluded participants with any concurrent disease or clinical psychopathology to constrain the search or to exclude illnesses associated with unintentional weight changes.[16 29] However, it is uncommon for an individual with overweight or obesity to be without any concurrent disease or clinical psychopathology due to the greatly increased risk of a wide range of comorbidities,[8] therefore, exclusion of these participants limits the representativeness of findings. It is considered beneficial to include participants with comorbid conditions where possible to maximise the generalisability of review findings.

To our knowledge, there is no up-to-date, comprehensive review investigating the effect of weight management interventions on a broad range of mental health outcomes in a representative sample of adults with overweight and obesity. Furthermore, no review has investigated the intervention components most supportive of mental health improvements. Understanding whether specific intervention components, such as psychological support, can attenuate the possible adverse effects to mental health is important for the development of future interventions. If data allows, this systematic review will apply subgroup analyses and meta-regression techniques to explore the differential effects of intervention or participant characteristics on mental health.

The conflicting findings of previous research and the absence of an up-to-date evaluation of the impact of weight loss interventions on mental health make it difficult to draw clear, reliable conclusions. A comprehensive updated review should increase understanding of the impact of weight management interventions on mental health. The most effective combination of intervention components should be investigated to facilitate improved decision making in intervention development, aiding the creation of an effective and supportive 'whole-person' intervention.

## Objectives

To assess the effectiveness of behavioural weight management interventions compared with minimal, inactive or 'standard care' control groups on mental health in adults with overweight and obesity.

Primary objective: (1) Quantify the effect of behavioural weight management interventions on mental health in adults with overweight and obesity.

Secondary objective: (2) Quantify if particular intervention or participant characteristics influence the effect of interventions on mental health.

## METHODS AND ANALYSIS

This systematic review protocol adhered to the Preferred Reporting Items for Systematic Reviews and Meta-Analyses Protocols (PRISMA) (online supplementary A).[30]

### Eligibility criteria

Studies will be selected according to the criteria outlined below:

#### Study designs

Original peer-reviewed primary research articles reporting randomised controlled trials (RCTs) or cluster RCTs will be included. No restrictions will be placed on year of publication.

#### Participants

Participants will be included if they are community-dwelling adults (≥18 years old with no upper age limit applied) with overweight or obesity (body mass index ≥25 kg/m$^2$) at baseline. Studies that include participants both under and over the age of 18 years will only be included if the data for participants 18 years and older is reported separately. Participants must be

seeking intentional weight loss through a behavioural programme. No restrictions will be made on participant demographics. To increase the generalisability of the findings to the general population with overweight and obesity, we will include studies that include people with comorbidities but we will exclude papers that focus exclusively on populations with a physical or mental comorbidity (eg, all participants have cancer), or pregnant women.

### Interventions
Studies will be included if they evaluated a behavioural weight management intervention that aims to achieve weight loss through changes in diet and/or physical activity. No restriction will be placed on intervention delivery duration, delivery format or on who delivers the intervention. Any study with multiple intervention arms will be included if at least one arm meets the inclusion criteria and separate results are presented for this arm. Interventions aiming to treat eating disorders or involving surgical and/or pharmacological intervention will be excluded.

### Comparators
Studies with a minimal/inactive/standard care control group will be included.

### Outcomes
Included studies are required to have measures of one or more of the following outcomes: quality of life; mood/affect; stress; self-esteem; body image; emotional eating; binge eating; depression; anxiety. These a priori defined outcomes were chosen as they were deemed to be the most relevant and frequently used in previous relevant literature.

### Timing
Defined outcomes must be measured and reported at preintervention and at minimum one follow-up point to be eligible for inclusion. The follow-up measurements closest to the time of intervention completion will be extracted for analysis to focus on the immediate intervention effects.

### Settings
Only studies involving participants living in community-based settings will be included.

### Language
Studies published in English language will be included. Non-English language publications will be excluded.

### Information sources and search strategy
The following databases were searched from inception to 7 May 2019:
- AMED.
- ASSIA.
- CINAHL.
- Cochrane database (CENTRAL).
- Embase.
- MEDLINE.
- PsycINFO.

Detailed search strategies for each electronic database were developed by RAJ, who has previous experience of conducting systematic reviews, with input from ERL, ALA and a medical librarian. The search strategy contains relevant key words and headings based on previous review articles[25 31–34] and is based on the concepts: (1) adults with overweight/obesity AND (2) weight management interventions AND (3) mental health outcomes AND (4) study designs. Terms were adapted from the MEDLINE search accordingly for each database (online supplementary B). The search was restricted to English-language papers, with no other restrictions applied. The search strategy was validated through consultation with the systematic review team.

### Other resource searches
To augment the results of the database search, the reference lists of included studies and previous relevant reviews will be searched.[9 16 24–27 29]

## Study records
### Data management and selection process
The search results were imported into Covidence systematic review software (Veritas Health Innovation, Melbourne, Australia), and duplications removed. Two researchers initially pilot screened an identical 500 articles to ensure consistency. Any discrepancies in the interpretation of the eligibility criteria were discussed between investigators, with a third reviewer assisting where necessary. On completion of pilot screening, the remaining title and abstracts will be independently screened for inclusion by two authors. The full text of articles identified as potentially relevant will be obtained and dually screened according to the eligibility criteria to ascertain the studies to be included in the review. Eligibility will be discussed for consensus between the two investigators, with a third investigator resolving discrepancies when required. Where necessary, we will seek additional information from study authors to resolve any questions about eligibility. Reasons for exclusion of articles at the full-text screening stage will be recorded. Reviewers will not be blinded to authors, institution or journal when screening articles.

Where studies are reported in more than one publication, all articles will be included and combined to make best use of the data. A PRISMA flow chart will be reported to show the process of study selection.[30]

### Data collection process
Studies meeting the inclusion criteria will have pertinent data extracted using a data extraction form. The data extraction form will be based on the Cochrane data extraction form (2011),[35] the Consolidated Standards of Reporting Trials statement (2010)[36] and the Cochrane Template for Intervention Description and Replication[37]

to ensure breadth and detail will be captured. The data extraction form will be pilot tested by two investigators on three studies to identify missing or superfluous data items. Independent data extraction will be completed by one investigator with full checking by a minimum of one further investigator. Discrepancies will be resolved through discussion, with use of a third investigator where necessary.

## Data items
Data to be extracted will include:
► General information (eg, study authors, publication year, country, funding source).
► Study details (eg, study aim, study design, randomisation method, blinding and allocation concealment).
► Participant information (eg, demographics, recruitment methods, sample size, comorbidities).
► Attrition/adherence (eg, total number of participants at baseline and follow-up measurements, differential attrition, attendance, study withdrawal, lost to follow-up).
► Intervention information (eg, setting, content, intervention duration and frequency, profession delivering the intervention, method of delivery, group or individual delivery).
► Comparator information (eg, setting, content, intervention duration and frequency, profession delivering the intervention, method of delivery, group or individual delivery).
► Outcomes (eg, mental health outcome(s) studied, whether self-reported or objectively measured, duration of follow-up, statistical analysis, intervention effect sizes).

If a study has multiple arms, data from any arm meeting the inclusion criteria will be extracted where possible. Study authors will be contacted if there are uncertainties regarding the study or missing data.

## Outcomes and prioritisation
For all outcomes, prioritisation will be given to units reported as raw data at baseline and postintervention over data presented as 'mean change' or equivalent. Where possible, data items will be extracted at both study and group level to permit analysis of overall and stratified data (eg, extracting stratified data to analyse moderation by sex). Study authors will be contacted to request any data required that is not available.

## Risk of bias in individual studies
Risk of bias (RoB) will be independently appraised by a minimum of two review authors. Discrepancies will be discussed between authors for a consensus and a third investigator will be consulted where required.

The Cochrane 'RoB' tool will be used to assess the RoB in the included studies.[38] The tool assesses the following study features as 'low risk', 'high risk' or 'unclear': (1) random sequence generation, (2) allocation concealment, (3) blinding of participants and personnel, (4)

blinding of outcome assessment, (5) incomplete outcome data and (6) selective reporting.

Other potential sources of bias not covered by the tool will be noted by review authors. Review authors will not be blinded to the included study's information (author names, journal of publication, affiliated institute). A RoB graph and summary table will be presented.

## Data synthesis
When the data permits, outcome data will be synthesised using a random-effects meta-analysis (Review Manager V.5.3, Cochrane Collaboration) due to the predicted diverse range of population and intervention types. Meta-analysis will be conducted on the outcome measures reported closest to the time of intervention completion, regardless of intervention duration, to focus analysis on the immediate intervention effects.

As it is likely a range of outcome measures will be identified, standardised mean difference (SMD) will be calculated. SMD will be categorised using thresholds as small (0.2), medium (0.5) and large (0.8).[39] Where possible, mean differences (for continuous data) and OR (for categorical data) and their 95% CIs will be calculated and reported.

### Sensitivity analysis
If considered useful after consultation with the review team, sensitivity analysis will be conducted to investigate the potential impact of RoB and participant characteristics on the effect estimates. The analysis will be restricted to different RoB levels to assess if study quality influences the effect estimates.

### Assessment of heterogeneity and reporting bias
Heterogeneity will be assessed using the $I^2$ statistic (and 95% CI). Heterogeneity will be categorised as low (0%–30%), moderate (30%–60%), substantial (60%–90%) and considerable (90%–100%).[40] In accordance with Cochrane recommendations, a funnel plot will be reported to assess the presence of publication bias.

### Analysis of subgroups or subsets
In the presence of sufficient data, subgroup analysis will compare:
► Population characteristics (eg, existing comorbidities, age, gender, degree of excess weight (overweight vs obese)).
► Intervention type (eg, diet vs exercise vs diet and exercise combination, including vs excluding psychological therapies).
► Intervention duration (eg, 1 day, 12 weeks, 52 weeks).
► Intervention delivery format (eg, face to face vs remote, individual vs group based).
► Comparator type (eg, intensities of comparator (minimal/inactive/standard care)).

If considered useful after consultation with the review team and in the presence of sufficient data on important covariates, meta-regression techniques will be

applied to identify and/or adjust for potential sources of heterogeneity.

## Narrative synthesis

Meta-analysis will be deemed inappropriate if significant heterogeneity is present or if we are unable to pool the outcomes. If meta-analysis is not possible, narrative synthesis and 'levels of evidence' assessment will be completed. This will be provided in the text and in a table format.

A ratings system, 'levels of evidence', will be used to draw conclusions of effectiveness. This will assess confidence in cumulative evidence at an outcome level. This is based on the methods applied by a previous review paper,[41] and is modified for the synthesis of RCTs only (online supplementary C). Included studies will be assessed on the level of evidence according to study quality and sample size. There are five possible levels of evidence ratings that can be achieved—strong, moderate, limited, inconclusive and no evidence for effect. Consistent positive findings in at least two thirds of studies is required to achieve 'strong', 'moderate' or 'limited' levels of evidence. In stratified analysis, we will assess study's levels of evidence according to intervention, participant or study characteristics. If meta-analysis is deemed inappropriate, we will graphically summarise our findings using harvest plots of extracted data.[42]

## Patient and public involvement

A lay summary of the proposed plan for the systematic review was shared with an established patient and public involvement (PPI) panel. The PPI panel gave feedback on the usefulness and relevance of the review aims and included outcomes. On review completion, the PPI panel will provide input on the lay summary of review findings and dissemination of findings.

## ETHICS AND DISSEMINATION

This systematic review will follow the PRISMA checklist. The completed systematic review will be disseminated in a peer-reviewed journal, at conferences and contribute towards the lead author's PhD thesis. The findings of the review will be of interest to participants of interventions, healthcare practitioners, policy-makers and researchers.

**Acknowledgements** The authors would like to thank the University of Cambridge School of Clinical Medicine librarian, Eleanor Barker, for assistance in developing the search strategy. We would like to thank the patient and public involvement (PPI) representatives who contributed to the development of this research.

**Contributors** RAJ conceived the study, designed the study, developed the initial search strategy and was responsible for drafting the manuscript. ALA conceived the study, participated in study design, development of the search strategy and reviewed drafts of the manuscript. ERL participated in study design, development of the search strategy and reviewed drafts of the manuscript. EMFvS and SJG contributed to the design of the study and reviewed drafts of the manuscript. All authors critically reviewed the manuscript and approved the final version submitted for publication.

**Funding** RAJ, ERL, ALA and SJG are supported by the Medical Research Council (MRC) (Grant MC_UU_12015/4). The University of Cambridge has received salary support in respect of SJG from the National Health Service in the East of England through the Clinical Academic Reserve. EMFvS is supported by the Medical Research Council (MRC) (Grant MC_UU_12015/7).

**Disclaimer** Funders had no involvement in study design, writing of the report or decision to submit the paper for publication.

**Competing interests** None declared.

**Patient consent for publication** Not required.

**Ethics approval** Ethical approval is not required as primary data will not be collected.

**Provenance and peer review** Not commissioned; externally peer reviewed.

**ORCID iD**
Rebecca A Jones http://orcid.org/0000-0003-2197-1175

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
