## [Reviewer comments · BMJ Open]

ARTICLE DETAILS

TITLE (PROVISIONAL)	The impact of adult weight management interventions on mental health: a systematic review and meta-analysis protocol.
AUTHORS	Jones, Rebecca A; Lawlor, Emma Ruth; Griffin, Simon J; van Sluijs, Esther; Ahern, Amy

VERSION 1 – REVIEW

REVIEWER	Laura Wilkinson Swansea University, UK.
REVIEW RETURNED	23-Sep-2019

GENERAL COMMENTS	This is an excellent thorough protocol. Clearly written and well planned. The authors notably take a very timely and progressive approach to the question at hand (e.g., contributing to the creation of 'whole person' interventions'). A few minor points for clarification should be considered by the authors below: 1. Page 6 lines 10 - 16: On the one hand the authors point out that a limitation of the previous literature has been the exclusion of participants based on 'concurrent disease or clinical psychopathology' and that this limits the representativeness. But on the other hand on page 7 lines 31-34, they mention that to increase generalizability that they will not include studies that focus only on populations with a physical or mental comorbidity. I think that this makes sense when you read it a few times - so studies should be included which haven't excluded based on comorbidities but shouldn't be exclusively restricted to that comorbidity? It might be worth an additional sentence or two to improve clarity. Page 7, line 17: Is there a rationale for not garnering any 'grey' literature? Page 12, Line 33, Do the authors have a particular threshold of heterogeneity that would be deemed 'significant' in mind and how would this relate to the categorisation laid out on Page 11?
--

REVIEWER	Flora Douglas Robert Gordon University Aberdeen, Scotland.
REVIEW RETURNED	07-Oct-2019

GENERAL COMMENTS	This paper is well written and clearly describes an important and overlooked area of public health research. I look forward to reading the findings paper when published.
---

REVIEWER	Jutta Mata University of Mannheim, Germany
REVIEW RETURNED	07-Nov-2019

GENERAL COMMENTS	This study protocol is very well-written, easy to understand and to the point. The research question is very relevant and a systematic review would be extremely relevant for the field. I have the following questions and suggestion to the authors: (1) Do you plan to differentiate between overweight and obesity (numerous studies show that health or mental health effects are only or more clearly pronounced in people with obesity than overweight)? (2) I would suggest searching additional data bases such as: Pubmed, PubPsych, Psyn dex, Web of Science, or PsyArxiv (3) The study protocol does not yet discuss how unpublished data will be considered. Options include also searching ProQuest Dissertation & Theses Global and writing to the mailing lists of relevant science organizations, asking for unpublished manuscripts. (4) The search terms used do not yet include controlled vocabulary (such as MeSH terms). This might further strengthen the search strategy.
--

VERSION 1 – AUTHOR RESPONSE

Reviewer: 1

Reviewer Name: Laura Wilkinson

Institution and Country: Swansea University, UK.

Please state any competing interests or state 'None declared': None declared.

Please leave your comments for the authors below

Reviewer: This is an excellent thorough protocol. Clearly written and well planned. The authors notably take a very timely and progressive approach to the question at hand (e.g., contributing to the creation of 'whole person' interventions'). A few minor points for clarification should be considered by the authors below:

Response: *Thank you very much for your complimentary comments on the protocol and the question posed.*

Reviewer: 1. Page 6 lines 10 - 16: On the one hand the authors point out that a limitation of the previous literature has been the exclusion of participants based on 'concurrent disease or clinical psychopathology' and that this limits the representativeness. But on the other hand on page 7 lines 31-34, they mention that to increase generalizability that they will not include studies that focus only on populations with a physical or mental comorbidity. I think that this makes sense when you read it a

few times - so studies should be included which haven't excluded based on comorbidities but shouldn't be exclusively restricted to that comorbidity? It might be worth an additional sentence or two to improve clarity.

Response: *We agree with the reviewer that this part of the manuscript could be confusing. The interpretation that the reviewer stated is correct – ‘studies should be included which haven't excluded based on comorbidities but shouldn't be exclusively restricted to that comorbidity’. Following the reviewer's comment and suggestion, we have added additional wording to the eligibility criteria on page 7 (subheading ‘Participants’) to aid clarity. Revised text: To increase the generalisability of the findings to the general population with overweight and obesity, we will include studies that include people with comorbidities but we will exclude papers that focus exclusively on populations with a physical or mental comorbidity (e.g. all participants have cancer), or pregnant women.*

Reviewer: Page 7, line 17: Is there a rationale for not garnering any 'grey' literature?

Response: *Within the review eligibility criteria, we have refined our included studies to peer-reviewed RCT research articles. This acts as quality threshold criteria to ensure that all included studies are of the highest quality. Furthermore, this criterion pragmatically restrains the search, especially with consideration of the large number of articles returned.*

Reviewer: Page 12, Line 33, Do the authors have a particular threshold of heterogeneity that would be deemed 'significant' in mind and how would this relate to the categorisation laid out on Page 11?

Response: *We would like to thank the reviewer for highlighting this. We will be considering more than just the chi-squared test for heterogeneity when assessing the suitability for meta-analysis. The Cochrane handbook identifies the limitations to the chi-squared test, and emphasises the caution that must be taken when interpreting the result (https://handbook-5-1.cochrane.org/chapter_9/9_5_2_identifying_and_measuring_heterogeneity.htm). Therefore, we will consider the I^2 statistic alongside thoroughly assessing the interventions, samples and measures in the studies, and ensuring not to combine those deemed to have significant heterogeneity when conducting meta-analyses. Decisions regarding assessment of heterogeneity and combining of studies will be reviewed by the full study team. Furthermore, we will conduct sensitivity analyses as we understand that some decisions may be deemed arbitrary/unclear – this is reflected in the Cochrane handbook recommendations (https://handbook-5-1.cochrane.org/chapter_9/9_7_sensitivity_analyses.htm).*

//

Reviewer: 2

Reviewer Name: Flora Douglas

Institution and Country: Robert Gordon University
Aberdeen, Scotland.

Please state any competing interests or state 'None declared': None declared.

Please leave your comments for the authors below

Reviewer: This paper is well written and clearly describes an important and overlooked area of public health research. I look forward to reading the findings paper when published.

Response: *Thank you very much for your positive appraisal of the protocol. We look forward to sharing the review findings in due course.*

//

Reviewer: 3

Reviewer Name: Jutta Mata

Institution and Country: University of Mannheim, Germany

Please state any competing interests or state 'None declared': None declared

Please leave your comments for the authors below

Reviewer: This study protocol is very well-written, easy to understand and to the point. The research question is very relevant and a systematic review would be extremely relevant for the field.

Response: *We would like to thank the reviewer for the positive assessment of the planned review.*

Reviewer: (1) Do you plan to differentiate between overweight and obesity (numerous studies show that health or mental health effects are only or more clearly pronounced in people with obesity than overweight)?

Response: *We agree with the reviewer that this would be interesting and important to explore. We hope to be able to differentiate by degree of excess weight through subgroup/subset analysis (stated on page 11, section 'Data analysis', subheading 'Analysis of subgroups or subsets'), however we are limited by the data available in included studies. With this in mind, we cannot determine at this time if this will be possible to quantitatively differentiate between overweight and obesity. If it is not possible to differentiate quantitatively then a narrative assessment will be sought, dependent on available information from the included studies.*

Reviewer: (2) I would suggest searching additional data bases such as: Pubmed, PubPsych, Psyndex, Web of Science, or PsyArxiv

Response: *We agree that is important to search in both psychological-, medical- and health-related databases, and therefore have ensured to search many various databases. We believe that applying an extensive search strategy to 7 databases of varying fields has comprehensively captured the existing relevant literature. Following submission of the manuscript the search and removal of duplicates has been completed, resulting in approximately 32,000 articles for title and abstract screening.*

Reviewer: (3) The study protocol does not yet discuss how unpublished data will be considered. Options include also searching ProQuest Dissertation & Theses Global and writing to the mailing lists of relevant science organizations, asking for unpublished manuscripts.

Response: *Within the review eligibility criteria, we have refined our included studies to peer-reviewed RCT research articles. This acts as quality threshold criteria to ensure that all included studies are of the highest quality. Furthermore, this criterion pragmatically restrains the search, especially with consideration of the large number of articles returned. We will contact authors where we have identified through our search that they have collected mental health data as part of a published RCT but have not published these outcomes yet.*

Reviewer: (4) The search terms used do not yet include controlled vocabulary (such as MeSH terms). This might further strengthen the search strategy.

Response: *We have included extensive MeSH terms in the Medline search, shown in Supplement B. We have used the 'explode' function for all MeSH terms to strengthen the search strategy further. We have ensured to translate these to the different databases used. The MeSH terms are shown in supplementary file B, and are listed below:*

- *exp Overweight/ or exp Obesity/*
- *exp Body Weight/ or exp Life Style/ or exp Physical Activity/ or exp Obesity Management/ or exp Diet Therapy/ or exp Exercise/ or exp Diet/ or exp Behavior Therapy/ or exp Health Education/*
- *exp Behavioral Symptoms/ or exp Emotions/ or exp Mental Disorders/ or exp Adaptation, Psychological/ or exp Mental Health/ or exp Quality of Life/ or exp Self Concept/*
- *exp Randomized Controlled Trials as Topic/*

VERSION 2 – REVIEW

REVIEWER	Jutta Mata University of Mannheim, Germany
REVIEW RETURNED	04-Dec-2019

GENERAL COMMENTS	The authors have addressed my concerns to my satisfaction. Good luck with your review!
---